# Mechanical Properties of High Carbon Low-Density Steels

**DOI:** 10.3390/ma16103852

**Published:** 2023-05-19

**Authors:** Jiří Hájek, Zbyšek Nový, Ludmila Kučerová, Hana Jirková, Črtomir Donik, Zdeněk Jansa

**Affiliations:** 1COMTES FHT a. s., 334 41 Dobrany, Czech Republic; zbysek.novy@comtesfht.cz; 2Regional Technological Institute, University of West Bohemia, 301 00 Plzen, Czech Republic; skal@rti.zcu.cz; 3Department of Materials and Engineering Metallurgy, University of West Bohemia, 301 00 Plzen, Czech Republic; hstankov@fst.zcu.cz; 4Institute of Metals and Technology (IMT), 1000 Ljubljana, Slovenia; 5New Technologies Research Centre, University of West Bohemia, 301 00 Plzen, Czech Republic; zjansa@ntc.zcu.cz

**Keywords:** low density steels, heat treatment, tensile test

## Abstract

The paper presents the possibilities of heat treatment of low-density structural steels usable for springs. Heats have been prepared with chemical compositions 0.7 wt% C and 1 wt% C, as well as 7 wt% Al and 5 wt% Al. Samples were prepared from ingots weighing approximately 50 kg. These ingots were homogenised, then forged, and hot rolled. Primary transformation temperatures and specific gravity values were determined for these alloys. For low-density steels, there usually needs to be a solution to achieve the required ductility values. At cooling rates of 50 °C/s and 100 °C/s, the kappa phase is not present. A SEM analysed the fracture surfaces for the presence of transit carbides during tempering. The martensite start temperatures ranged from 55–131 °C, depending on the chemical composition. The densities of the measured alloys were 7.08 g/cm^3^ and 7.18 g/cm^3^, respectively. Therefore, heat treatment variation was carried out to achieve a tensile strength of over 2500 MPa, with ductility of almost 4%. Hardnesses above 60 HRC were achieved for 1 wt% C heats using the appropriate heat treatment.

## 1. Introduction

Low-density steels are an emerging class of structural materials for applications mainly in the automotive, chemical, and aerospace industries [1,2]. With the growing demand for optimising fuel consumption and the ever-tightening regulations dealing with CO_2_ emissions, the automotive industry is increasingly emphasising the need to reduce product weight while maintaining existing qualities, especially concerning occupant safety. According to a study by Ivan Gutierrez-Urrutia [3], reducing the weight of a vehicle by 100 kg reduces CO_2_ emissions by approximately 8.5 g per km. Various strategies are being adopted to reduce the weight of cars, such as structural modifications to remove redundant materials or replacing materials with better properties. High-strength steel, which provides the same mechanical properties while using thinner walls, is often used as a substitute for conventional steel. Another option is to use a lower-density material, which reduces the product’s weight while maintaining the material’s volume [4]. Such materials, today, include lightweight alloys, such as aluminium–magnesium alloys and various composites. However, these materials only sometimes meet the high mechanical and thermal stress requirements. Therefore, research nowadays focuses on developing lower-weight steels with high strength and low density [3,5]. 

The principle concept of alloying in low-density steels is simple, but the metallurgical feed technology or subsequent heat treatment is complicated. The addition of Al to the Fe–C system has a large effect on the phase fields and the phase constituent [6]. The ranges of composition and temperature for the high-temperature peritectic transformation are enlarged, indicating that a higher level of elemental segregation (micro and macro) can occur during the solidification process. The (δ + γ) area is extended, and the single γ phase area is shifted to the right as the C concentrations of the “S point and the E point are increased. A new phase, called κ-carbide, is introduced when the Al content is higher than 2 wt%. The (γ + M_3_C) area in the Fe– system is replaced by (γ + κ) and (γ + κ + M_3_C) areas [7].

As mentioned above, the addition of aluminium to steels results in the stabilisation of the ferrite. Another problem is the occurrence of unsuitable intermetallics, which increase in frequency as the aluminium content of the steel increases. Their amount increases significantly when exceeding 6.5 wt% Al. The solubility of aluminium in a solid solution is limited. A detailed description of the occurrence of intermetallics in the Fe-Al system is described in [8]. In order to be able to use Fe-Al-based alloys in practice, it is necessary to add a suitable amount of austenite-forming alloying elements, which are Mn or, more effectively, C. EN 100Cr6-based steels, which contain 1% C and 1.5% Cr. They seem to be the most promising in this respect. In the case of alloying concepts based on Fe-15Mn-10Al-0.8C, specific strengths up to 200 MPa cm^3^/g can be achieved [9].

The large Mn and Al contents in steel are difficult to process during the production process of Fe-Mn-Al-C steels [10]. Intense chemical reactions can occur between the melt and refractories, leading to impurities and heterogeneities in the chemical composition [1]. During the melting and solidification process, Mn and Al also react with atmospheric oxygen, nitrogen, and sulphur, forming AlN, MnS, and a variety of other heavy oxide inclusions [11,12]. As a result, various undesirable phenomena can occur during steel casting, such as forming surface defects or cracks, elimination of brittle phases, and decarburisation. In order to overcome these difficulties, new casting methods must be introduced. In addition, the microstructure arrangement after casting and its modification by heat treatment must be addressed [1]. Most of the existing concepts are aimed at something other than use in structural parts requiring high loads. This can only be achieved by appropriate hardening and tempering [13]. 

Creating and describing Fe-Mn-Al-C and Fe-Al-C phase diagrams is a challenging task. Only in the last decade have new phases and phase transformations been identified in this quaternary system’s alloys [14]. In general, the resulting structure can consist of equilibrium and non-equilibrium phases. The matrix in these steels can be either ferrite, austenite, or a mixture of these. As already mentioned, aluminium is a strong ferrite-forming element. Therefore, ferrite is very stable at higher contents of this element. However, the stability of ferrite is undesirable for the future modification of mechanical properties by heat treatment. It is, therefore, necessary to compensate for the high alumina content with austenite-forming elements, such as Mn and C. In this way, a fully austenitic structure can be achieved, at least, at higher temperatures, thus achieving suitable sub-conditions for heat treatment [3,15].

In our experiment, the chemical composition most closely resembles the alloying concepts of Fe-5Al-1C and Fe-6Al-1C. The work focuses on the description of the essential transformation reactions. Virtually all of these steels, after rolling, contain a microstructure consisting of ferrite, predominantly lamellar pearlite, and kappa carbides. The authors precisely describe the individual phases or mixtures in [9]. Our work builds on the above-cited works and comes up with an alloying concept that could be successfully applied in production. The most significant use of the proposed steel is for parts operating under high dynamic stress, e.g., connecting rods, pistons, etc. High thermal stability will also be an essential parameter in some similar applications. Our concept assumes the high thermal stability of the structure combined with heat treatment. Another benefit will be higher corrosion resistance, which is inherently typical for aluminium alloy steels [15].

In our paper, two types of steels are presented, which are based on the alloying concepts presented in [15,16,17] and modified so that their heat treatment can be carried out efficiently to achieve the required strength and deformation characteristics. The aim is to achieve a yield strength to ultimate strength ratio of max. 0.85 at an Rm of at least 1500 MPa and a ductility of at least 5%. This paper aims to demonstrate the effect of the Al:C ratio on the mechanical properties of steels with increased aluminium content and the ability of phase transformation during heat treatment.

## 2. Materials and Methods

### 2.1. Design of the Experiment

JMatPro software version 12.1 (Sente Software Ltd., Guildford, UK) was used, in combination with analytical techniques for experimental planning, to design a suitable chemical composition for the low-density steels [18]. Variants of the chemical compositions of steels with reduced density were selected using JMatPro supplemented by the Design of the Experiment (DOE) method. Due to the weight requirement, the minimum weight for given energy storage (*M*_2_) was used to calculate the Ashby coefficient. The coefficient is based on [19,20]:$$M_{2}=\sigma_{f}^{2}/E\rho$$
where *σ_f_* is the yield stress of the given material, *E* is Young’s modulus, and *ρ* is the density of the material. The variants marked 7Al and 5Al were designed on the basis of these calculations. The chemical compositions of both cast heats were determined using a Q4 TASMAN optical emission spectrometer (Bruker AXS GmbH, Karlsruhe, Germany) (Table 1).

### 2.2. Casting and Hot Rolling of Experimental Steels

The production of the experimental material was carried out in a vacuum (rough vacuum) induction furnace model TN—00-361, Prvni Zelezarska Kladno, Czech Republic in COMTES FHT. The experimental heats were prepared by pouring into casts with dimensions of D = 110 mm, L = 500 mm. The ingots prepared in this way had a mass of 50 kg (Figure 1). The melting and casting technology of “high-aluminium” steel is characterised by an excess of the deoxidising agent, aluminium, so all of the oxygen in the steel is bound. For this reason, there is no need to use other deoxidising agents. Massive deoxidation with aluminium produces a large amount of slag, which needs to be collected during the melting process. Furthermore, the process is characterised by the bubbling of argon, which both promotes the leaching of slag and binds hydrogen and nitrogen. Argon was used as a protective atmosphere.

After cooling, homogenisation annealing was included. It was annealed for 8 h at 1130 °C. Heating was carried out simultaneously in the furnace, with a 2 h delay, at 550 °C and 870 °C. Cooling was performed in the furnace.

The ingots were forged at 1180 °C, with a lower forged temperature of 980 °C. The upset reduction was 2.5.

Forged ingots were then hot rolled at 1180 °C. The rolling was carried out, at 20% reduction, to a final sheet thickness of 14 mm. Samples were prepared from these sheets for subsequent experiments (Figure 1).

### 2.3. Materials Analysis

The samples for metallographic experiments were prepared by conventional mechanical grinding. This was followed by polishing on cloths, with a diamond suspension in steps of 9, 3, and 1 μm. The structure was etched with 3% Nital. (Lach-Ner, s.r.o., Neratovice, Czech Republic). Microstructures were documented using an Olympus BX 61 light microscope (Olympus, Shinjuku, Tokyo, Japan) and a Crossbeam Auriga electron microscope with an FEG cathode (Zeiss, Oberkochen, Germany).

A push rod dilatometer L78 RITA (Rapid Inductive Thermal Analysis, produced by Linseis, Selb, Germany) was used for the detection and analysis of phase transformations in the steels, as well as for the preparation of the microstructure by simulating temperature changes in any location of the heat treated body. Cylindrical specimens of 3–5 mm diameter and 10 mm length are heated in an induction coil and cooled by gas (helium, argon, nitrogen).

Tensile testing was carried out according to EN ISO 6892-1 [21]. There were three specimens processed with the appropriate heat treatment that were evaluated each time. The dimensions of the samples can be seen in Figure 2. The measurements were tested on an electromechanical testing machine—a Zwick/Roell 250 kN laser extensometer.

## 3. Results

### 3.1. Investigations of the Microstructure after Casting and Hot Rolling

Samples were cut from the base of the ingots for metallographic analysis. As a result of significant segregations during solidification, the structure of the heat 7Al (0.7% C and 7% Al) presented a characteristic dendritic arrangement in the as-cast state. This can be observed in Figure 3a, where the bright spots corresponding to the ferrite phase can mainly be seen; the latter forms the major and minor axes of the dendrites in the structure. The brown areas correspond to the eutectoid mixture, pearlite, which fills the interdendritic spaces. Figure 3b shows the morphology of the eutectoid mixture (P-pearlite). Carbides are excluded, here, in both lamellar and globular forms. The kappa phase (κ) is excluded along the boundaries of the original austenitic grains. These are allotriomorphic particles formed at grain boundaries. These particles nucleate, mainly, at the grain boundaries of the original austenite and grow further along these boundaries. However, the kappa phase also occurs in the structure, as plates nucleate at grain boundaries and grow inside the grains. In some cases, the kappa phase occurs independently inside the ferrite grains as idiomorphic crystals. The cooling kinetics or local chemical composition determines the distribution and morphology of the particles.

In the case of the heat 5Al (1% C, 5% Al), the structure is fully pearlitic after casting (Figure 4a,b). The morphology of the pearlite is predominantly lamellar. A kappa phase can also be observed locally at the boundaries of the original austenitic grains. In this case, however, its abundance in the structure is much lower. Some carbon is excluded from the structure in the form of secondary chromium carbides with globular morphologies. If the %C and %Al ratio is sufficient, ferrite can be completely eliminated. The specific proportion will, of course, always depend on variables in the production technology. However, the experiments carried out so far show that, at a ratio of %C:%Al = 1:5, it is possible to leave the two-phase structure of ferrite and austenite under normal casting conditions.

### 3.2. Thermo-Physical Measurements

#### 3.2.1. Determination of Transformation Temperatures


Heats 0.7% C and 7% Al


Dilatometric measurements were performed to properly design the heat treatment parameters and understand the phase transformations in the designed steels. The first dilatometric measurement was performed with a heating rate of 3 °C/min to a temperature of 1200 °C. This is a standard rate that is suitable for determining critical points (Figure 5).

For the heat 7Al (0.7% C, 5% Al), a pronounced Currie point at about 625 °C is evident. Furthermore, an increase in volume can be observed at 740 °C. This can be linked to the decomposition of cementite. The decrease in volume associated with austenitisation starts at 843 °C. Decomposition of the kappa phase also occurs from 860 °C, resulting in an increase in volume. Austenitisation and carbide dissolution occur simultaneously, and they also have opposite effects on the volume change. It is, therefore, difficult to observe the exact end of the kappa phase decomposition or dissolution of other carbides by observing dilatometric indications. However, it can be concluded that, above 910 °C, the dissolution of the kappa phase is complete and continues only with further stages of austenitisation.

In Figure 6, it can be seen that the structure formed during dilatometric measurements, when the sample was cooled, immediately after reaching the appropriate temperature at a rate exceeding 100 °C/s. Up to a heating temperature of 840 °C, the structure is composed of pearlite, ferrite, and kappa phases. Only the morphology of the pearlite changes. At 870 °C, austenitisation occurs in the pearlite region. This confirms the results of the dilatometric measurements, which showed the dissolution of the kappa phase in the temperature range from 860 °C–910 °C. The images show a gradual decrease in the amount of this phase throughout the volume. Globular small cementite particles are the last to dissolve. Their dissolution ends at about 960 °C. Furthermore, the microstructure consists only of the two-phase structure of austenite and ferrite. Table 2 shows the varying proportion of ferrite as a function of temperature.

In the description of the structures, here, we are talking about the γ phase—austenite. Of course, after cooling, the figures show areas that transformed into martensite with retained austenite. At the same time, apparently, with the increasing temperature, there is further formation of austenite at the expense of the ferritic phase (compare the microstructures at 930 °C and 1100 °C).

Figure 7 shows the SEM images of samples quenched from temperatures of 840 °C and 960 °C. It was confirmed that, at 840 °C, the structure consists of pearlite, delta ferrite, and kappa phase, occurring mainly at the boundaries of the original austenitic grains. The kappa phase naturally contains a higher Al content than ferrite. It is clear, from the analyses of the phases (Table 3) occurring at 960 °C, that there is no major redistribution of alloying elements between ferrite and austenite.


Heat 5Al (1.0 % C and 5% Al)


The same dilatometric measurement was performed for the heat with the higher aluminium content to determine the individual transformation temperatures with a heating rate of 3 °C/min (Figure 8). In this case, the Currie point temperature can also be observed. When the temperature exceeds 800 °C, the decomposition of the carbide phases occurs first, followed by the formation of the first austenite grains (approx. 810 °C). As in the previous case, it is difficult to observe the exact temperature of the dissolution of the carbide phases with dilatometric measurement.

In the images of the structures formed during quenching from different temperatures during dilatometric measurements, it is clear that no phase transformations occur in the structure up to 800 °C (Figure 9). At this temperature, the kappa phase starts to decompose. This is followed by the transformation of pearlite into austenite. At 900 °C, the pearlite mixture is already fully transformed. With a subsequent increase in temperature, only a gradual dissolution of the secondary chromium carbides into austenite occurs.

Figure 10 shows SEM images at 810 °C and 900 °C. The image on the right (810 °C) shows the formation of the first austenitic grains. At the same time, globular chromium carbides and lamellar pearlite are visible in the structure. There is a clear correspondence between the chemical composition of the austenite and the ferritic matrix (Table 4). At 900 °C, isolated grains of ferrite are present in the structure, with the austenitic phase dominating (transformed to martensite in the image). Part of the chromium carbide is dissolved. As already mentioned, the chemical composition does not differ significantly between the phases. The aluminium content, in this case, is slightly higher in the delta ferrite by about 1 wt%.

#### 3.2.2. Dilatometric Measurements at Cooling

Another important piece of information was the determination of the temperature of the martensite start. Based on previous dilatometric analyses, austenitisation temperatures of 930 °C and 900 °C were selected (see Table 5). As can be seen in Figure 11, the determination of the martensite start is based on the evaluation of the transformation temperature, as 1% and 99% of the change of the elongation dependence, from austenite to transformation products. The austenitisation parameters are shown in Table 5.

The temperatures at the start of the martensitic transformation were determined based on these measurements. The Ms temperature is somewhat dependent on the cooling rate, so measurements were made at 50 °C/s and 100 °C/s (Table 6). The measurements were always performed twice for each heat or cooling rate.

The results show a significant influence of the structure’s aluminium content on the Ms temperature. It is evident that, with an increasing rate, the temperature of Ms increases by about 20 °C.

#### 3.2.3. Dilatometric Measurements during Tempering

These steels will be used in the quenched and tempered states. Therefore, dilatometric measurements were also carried out during anisothermal tempering. Austenitisation was carried out at 930 °C for heat 7Al (0.7% C, 7% Al) and 900 °C for heat 5Al (1% C and 5% Al), following the previous experiments. In both cases, the austenitisation time was 20 min. The cooling rate from the austenitisation temperature was 100 °C/s. Subsequent tempering was carried out at a heating rate of 3 °C/min.

In the case of Heat 7Al (0.7% C, 7% Al), significant carbon precipitation from the solid solution and the formation of transit carbides was detected at temperatures above 100 °C. Significant carbide coarsening and cementite formation occurred above 550 °C (Figure 12).

SEM analysis is an essential complement to dilatometric analysis in the case of tempering (Figure 13 and Figure 14). The tempering process was interrupted at the specified temperature, and cooling was included at a rate of about 200 °C/s. Subsequently, a metallographic section was prepared.

From the image taken after the heat to 250 °C and the subsequent cooling, it is clear that transit carbides have already formed during the decay of the original martensitic plates. Retained austenite (RA) is also visible in the image. Its increased abundance can be observed close to the ferritic grains. The carbides are plate-like in morphology. Their length does not exceed 1 μm in most cases. They show considerable orderliness, with their growth related to the orientation of the crystal lattice matrix. When the tempering temperature is raised to 350 °C, there is further leaching of carbon from the supersaturated solid solution and the formation of additional transit carbides with platelet morphology.

Their size also remains unchanged. At 350 °C, the amount of RA remains unchanged. A further increase to 450 °C shows a coarsening of the transit carbides. Still, 450 °C is insufficient for the disintegration of retained austenite. No carbide precipitation occurs in the ferrite grains formed during solidification itself. It is only at temperatures above 500 °C that the seed changes occur. The 550 °C image already shows the residual austenite transformation into a mixture of ferrite and carbides with plate-like morphology. Coagulation of the carbides also occurs. At the same time, very fine carbides are precipitated within the ferritic grains. The structure formed at 650 °C consists of massive carbides exceeding 5 μm. New fine carbides are also precipitated. Carbides are excluded within the original ferritic grains.

In the case of 5Al heat (1% C and 5% Al), the structure shows lower stability after quenching (Figure 15). The formation of transit carbides occurs at temperatures about 50 °C lower than in the previous heats 7Al. The same applies to the formation of cementite. It is, therefore, evident that the lower the aluminium content or the higher the carbon content of the solid solution, the lower the temperature of the post-shift decay of RA and the precipitation of carbides at lower temperatures.

Dilatometric analysis was complemented by the metallographic analysis of martensite decays by SEM (Figure 16 and Figure 17).

The microstructure analysis shows that the amount of transit carbides formed due to martensite decomposition is much higher for heat 5Al than for heat 7Al.

Consistent with the previous heat, we can say that it starts to form at temperatures below 200 °C, following the dilatometric indications (Figure 15). The morphology of the carbides is mainly plate-like. As the tempering temperature increases to 350 °C, the carbides do not coagulate, but they increase slightly in quantity. The coarsening of the carbides occurs only at 450 °C (Figure 16). The plate-like morphology is preserved. When the temperature increases above 550 °C, the transit carbides are already coarsened, and cementite particles are formed. The BED images show that the stability of austenite decreases significantly from tempering temperatures of about 500 °C.

### 3.3. Heat Treatment

First, the 7Al (0.7% C, 7% Al) heat was processed. The heat treatment conditions are given in Table 7.

For heat 5Al (0.5% C, 5% Al), the quenching and tempering conditions were adjusted according to Table 8. All samples were quenched from the austenitising temperature in oil.

After heat treatment, hardness measurements were performed on metallographic cross-sections. The hardness is significantly higher for heat 5Al (5% Al, 1% C). This is probably due to the material’s ability to achieve homogeneous austenite with a subsequent martensitic transformation. The hardness curves also show the structure’s high stability. The hardness remains practically unchanged in the case of 5Al (1% C, 5% Al) up to 450 °C; then, it drops sharply.

In the case of heat 7Al (0.7% C, 7% Al), the stability of the microstructure is even higher. The hardness decreases at temperatures above 500 °C (Figure 18). The microstructure images confirm these measurements in Figure 13 and Figure 14.

#### Mechanical Properties after Heat Treatment

The mechanical properties were determined by tensile testing for selected heat treatment parameters based on results from previous analyses. Table 9 shows the heat treatment schedule for tensile test specimens.

Heat 7Al exhibits a maximum strength of over 1600 MPa (Figure 19, Table 10) at an elongation of approximately 11%. The highest strength is obtained after quenching and tempering to 300 °C. The lowest elongation is achieved when quenched and tempered to 550 °C. A significant decrease in strength occurs after tempering at 675 °C. Combining quenching and tempering at 425 °C achieves the highest tensile strength. With this combination of parameters, the strength equals 1.5 GPa, and the elongation is 14.2%.

Heat 5Al has a maximum strength of 2500 MPa at tempering temperatures of 275 °C and 400 °C (Figure 20, Table 11). Elongations are below 3%. The tensile strength and yield strength decrease from a tempering temperature of 525 °C when the ultimate strength and yield strength drop to 1864 MPa and 1800 MPa, respectively. The increase in ductility was only found at a tempering temperature of 650 °C at an ultimate strength of 1372 MPa. The ratio of yield strength to tensile strength is approximately 0.85.

Some exceptions are the strength and deformation characteristics, in the case of tempering at 525 °C, when the toughness of the steel decreases. This is probably due to the excluded cementite at the boundary of the original martensitic plates. This phenomenon was detected in Figure 16 and Figure 17 with the ‘dilatometry tempering’ at 550 °C (indicated by the red arrow). The samples exhibit the highest ductility and the lowest strength at the tempering temperature of 650 °C, when the cementite particles are spheroidised. It can be assumed from the microstructure images that the ferrite has not yet recrystallised, and it still retains the relief corresponding to the previous martensitic grains.

## 4. Discussion

### 4.1. Metallography and Alloying Concept

Based on the metallographic research of the heats, it is obvious how crucial the ratio wt% of carbon to aluminium is for leaving delta ferrite in the structure. This ratio is obviously influenced by the nickel and manganese contents, but their influence is marginal. This is illustrated in Figure 21. Here, the effect of elemental content, as a function of increasing austenite content in the structure at 1000 °C, is expressed based on a calculation in JMatPro.

Based on the experiments, it is evident that the proportion of carbon and aluminium required to leave delta ferrite in the structure is at least 1:5.

### 4.2. Dilatometry

Based on dilatometric experiments, the Currie point was lower than that of standard structural steel. In the case of both heats, the Currie point was practically identical (625 °C for Heat 7Al, 624.8 °C for Heat 5Al). In heat 7Al, the two-phase structure of austenite and ferrite is not abandoned until 1200 °C. However, from the metallographic analysis, it is clear that the amount of ferrite gradually decreases with an increasing temperature. The minimum is at 1200 °C. With a further increase in temperature, there is no further decrease in the amount of ferrite in the structure.

Furthermore, it is possible to determine the onset of austenitisation for heat 7Al (0.7% C and 7% Al) at 843 °C. Decomposition of the kappa phase takes place between 860 °C and 910 °C. Based on these measurements, the recommended austenitising temperature for industrial use is 930 °C.

For the heat 5Al (1% C and 5% Al), austenitisation starts at lower temperatures. The whole austenitisation process takes place from 810 °C to 900 °C. In this case, the recommended austenitising temperature is 900 °C.

From the analysis of both heats, the kappa phase does not occur in the structures at temperatures above 910 °C. The Kappa phase also contains the highest amount of aluminium (6.4% in the case of heat 7Al).

The results of dilatometric measurements during hardening show a significant influence of aluminium content in the heat on the temperature Ms.

The results confirmed the conclusions of [15] in these heats. Increasing the cooling rate affected the temperatures of the martensite start. The reason is that, at lower cooling rates, the carbon atoms in the solid solution segregate at the sites of the defects and interact with the dislocations, thus forming whole clusters of carbon atoms. At low cooling rates, the clusters of carbon atoms increase the strength of the austenite, leading to a decrease in the martensite start temperature. At high cooling rates, the reported clusters of carbon atoms hardly form and do not strengthen the solid solution.

From dilatometric experiments, which monitored the behaviour of the steel during tempering in the temperature range from 200 °C–700 °C, it is evident that the structures are relatively stable and resistant to tempering. In the case of heat 7Al, the hardness drops only at temperatures above 500 °C, as documented by SEM analysis. In the structure, tempering to 250 °C produces fine carbides with plate-like characteristics. These structures are virtually unchanged until the temperature reaches 500 °C. Above 500 °C, the residual austenite decomposes into a mixture of ferrite and fine carbides. The coarsening and spheroidisation of the carbides, as well as the formation of cementite, occur as well.

### 4.3. Hardness after Quenching and Tempering

Heat 5Al has a lower resistance against tempering. The hardness drop occurs from 450 °C upwards. Above this temperature, retained austenite breaks down. The coarsening of the carbides’ formation of cementite also occurs.

### 4.4. Tensile Test

From the tensile test of heat 7Al, it is clear that the highest strength is achieved at tempering temperatures of 350 °C, when the formation of fine carbides at the twin interface is evident in the microstructure. The retained austenite is still present in the structure. A further increase in the tempering temperature to 425 °C results in a strength decrease and increased ductility. It is only due to the coarsening of the carbides formed in the previous tempering stages. If quenching is followed by tempering to 550 °C, there is a significant decrease in ductility. It can be associated with decomposing residual austenite into a mixture of ferrite and carbides. The significant decrease in strength is associated with quenching and tempering to 675 °C, when significant migration and coagulation of the carbides occurs. In the case of heat 5Al, in terms of strength and deformation characteristics, the highest strengths are at the tempering temperatures of 275 °C and 400 °C, respectively, when there is a high occurrence of metastable carbides with plate-like characteristics in the structure. At a tempering temperature of 525 °C, there is a significant decrease in ductility. This is attributed to the break-up of residual austenite and the formation of areas with metastable carbides. On the other hand, the cementitious particles gradually form spheroids at this temperature. The high ductility, after tempering at 650 °C, is related to the coagulation of the cementite.

The ranges of tensile strength and total elongation of some of the low-density steels are shown in the so-called “banana diagram” in Figure 22 [22].

## 5. Conclusions

The results of the experiments can be summarised in the following points:–The key to leaving delta ferrite in the structure is the ratio of carbon and aluminium content. This ratio is also influenced by the nickel and manganese content, but their influence is marginal.–In the case of heat 7Al, the two-phase structure of austenite and ferrite is not abandoned below 1200 °C. However, from metallographic analysis, it is clear that the amount of ferrite gradually decreases with the increasing temperature. A minimum is observed at 1200 °C. With a further increase in temperature, there is no further decrease in the amount of ferrite in the structure.

From the analyses of both heats, the kappa phase does not occur at temperatures above 910 °C.
–Evidently, increasing the cooling rate affected the temperatures of martensite start.–From dilatometric experiments that monitored the behaviour of the steels during tempering in the temperature range from 200 °C–700 °C, it is evident that the structures are relatively stable and resistant to tempering.–The steel referred to in the paper as 7Al reaches an ultimate strength of over 1600 MPa, with a ductility of over 10%. Steel 5Al reaches a tensile strength of over 2500 MPa. Obviously, if the appropriate Al:C ratio is exceeded and the proportion of delta ferrite remains in the structure, the strength of the steel decreases as the proportion of ferrite increases. Maximum strengths cannot be achieved in the case of duplex or triplex low-density steels.–The reduction in specific gravity was almost 10% for the heat with 7 wt% Al. In the case of heat with 5 wt% Al, the reduction was about 9% in specific gravity.

The table below (Table 12) gives the most important results

## Figures and Tables

**Figure 1 materials-16-03852-f001:**
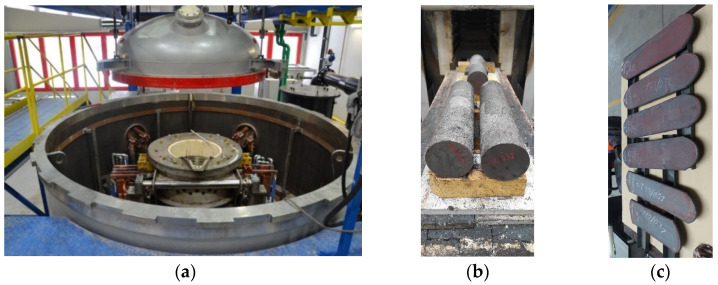
The preparation of experimental material: (**a**) vacuum induction furnace; (**b**) cast ingots prepared for homogenisation annealing; (**c**) final form of the sheets.

**Figure 2 materials-16-03852-f002:**
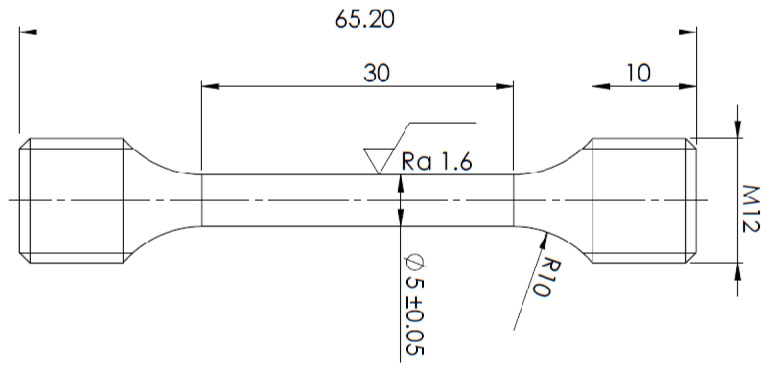
The specimen for tensile testing, according to EN ISO 6892-1.

**Figure 3 materials-16-03852-f003:**
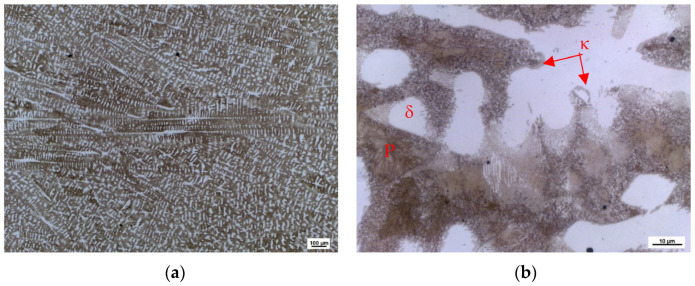
Heat 7Al, ingots’ bottoms: (**a**) magnification 50×; (**b**) magnification 500×.

**Figure 4 materials-16-03852-f004:**
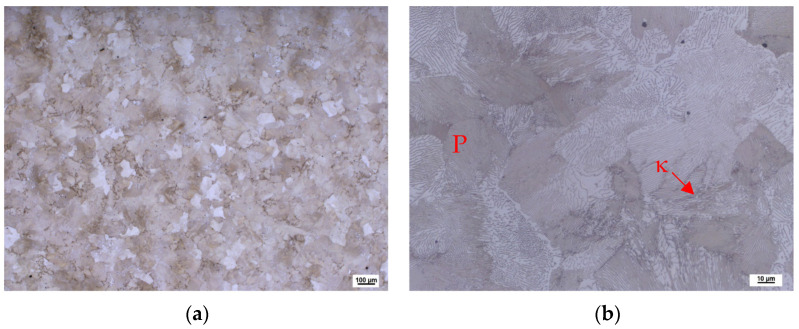
Heat 5Al, ingots’ bottoms: (**a**) magnification 50×; (**b**) magnification 500×.

**Figure 5 materials-16-03852-f005:**
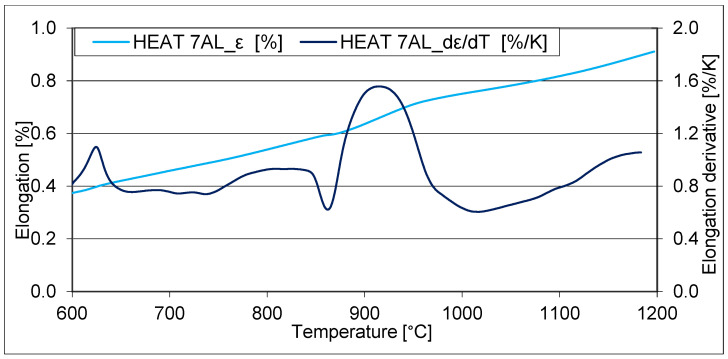
The dilatometric curve and its derivation—heating up at 3 °C/min. Heat 7Al (0.7% C and 7% Al).

**Figure 6 materials-16-03852-f006:**
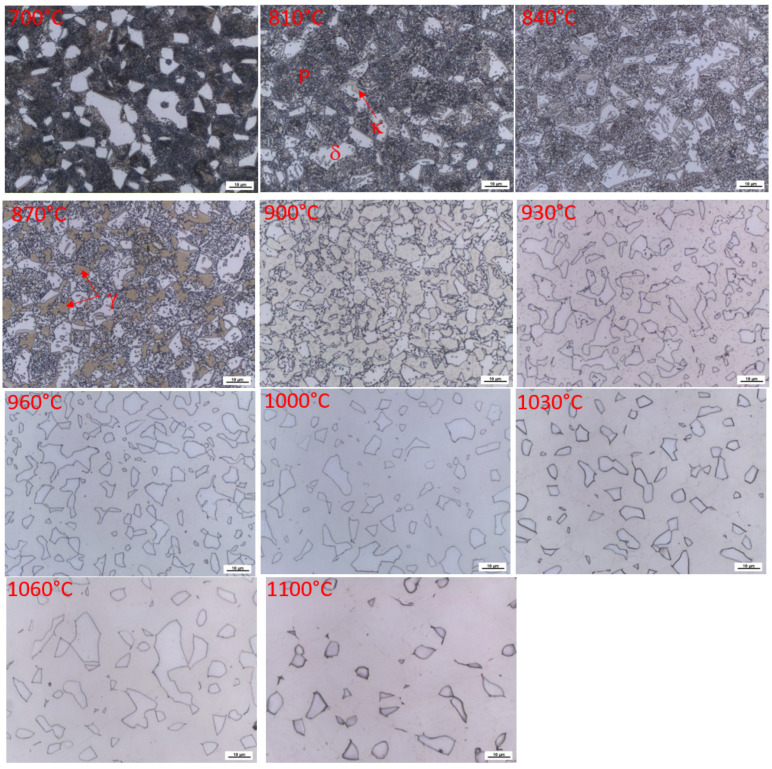
The microstructure of samples after dilatometric measurements were interrupted at different austenitisation temperatures, Heat 7Al (0.7 wt% C and 7 wt% Al). Magnified 1000× SEM observations, including the EDS analyses of individual phases, were also performed to confirm the previous statements.

**Figure 7 materials-16-03852-f007:**
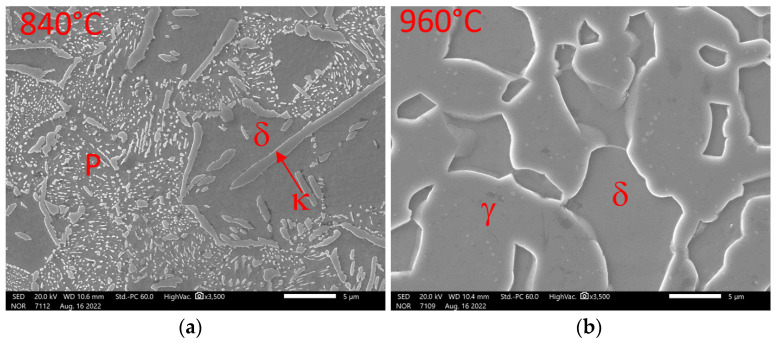
Detail of structures formed during the hardening of the samples in the dilatometer. Heat 7Al (0.7% C, 7% Al). SEM: (**a**) structural composition on cooling from 840 °C; (**b**) structural composition on cooling from 960 °C.

**Figure 8 materials-16-03852-f008:**
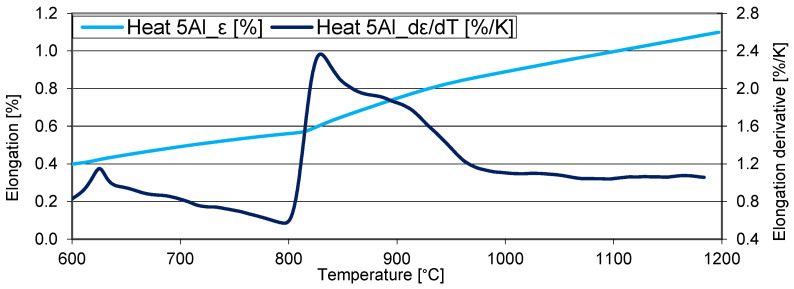
Dilatometric curve—heating at 3 °C/min. Heat 5Al (1% C and 5% Al).

**Figure 9 materials-16-03852-f009:**
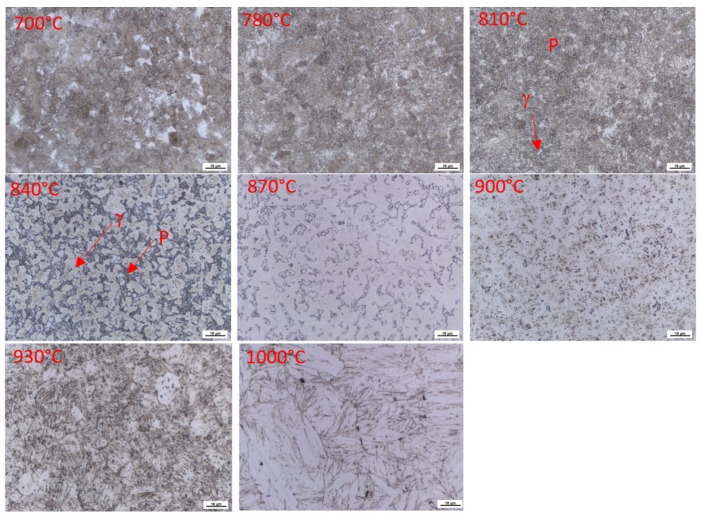
The microstructure of samples after dilatometric measurements, interrupted at different austenitisation temperatures, Heat 5Al (1.0% C and 5 wt% Al). Magnified 1000×.

**Figure 10 materials-16-03852-f010:**
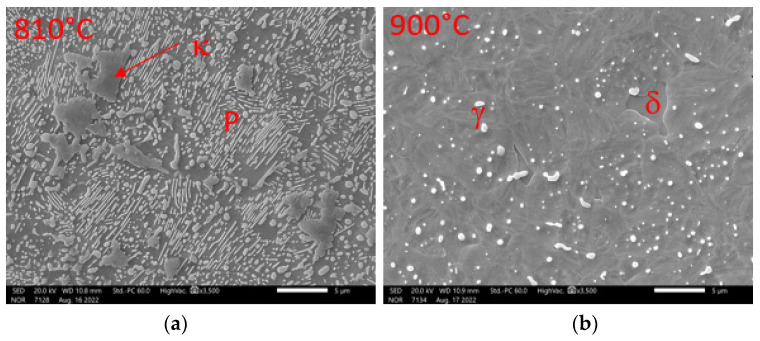
Detail of microstructures formed during dilatometric measurements. Heat 7Al (0.7% C, 7% Al). SEM Magnification 3500×; (**a**) austenitisation temperature 810 °C; (**b**) austenitisation temperature 900 °C.

**Figure 11 materials-16-03852-f011:**
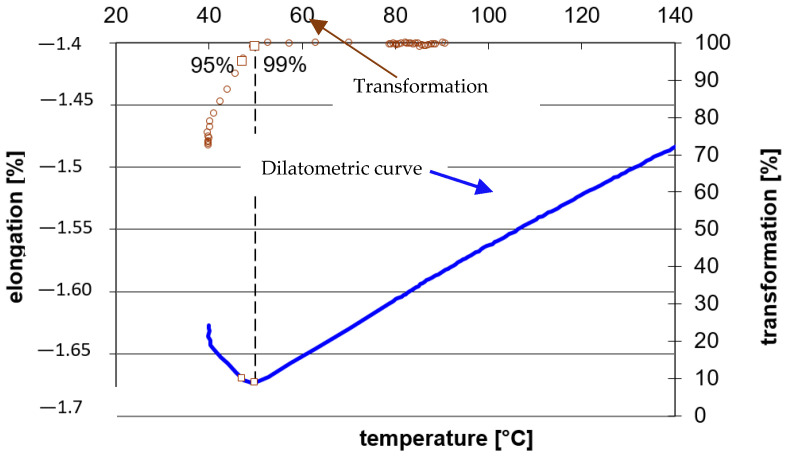
The determination of martensite start temperature Heat 7Al—cooling rate 50 °C/s.

**Figure 12 materials-16-03852-f012:**
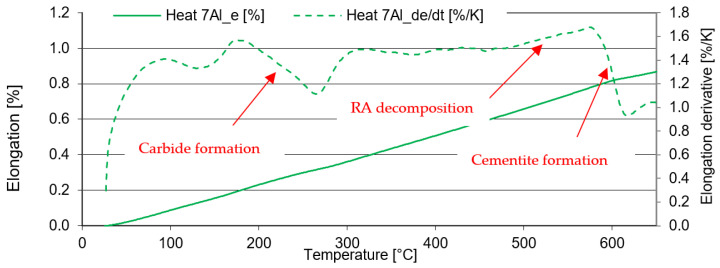
Tempering dilatometric curve—heating up at 3 °C/min. Heat 7Al.

**Figure 13 materials-16-03852-f013:**
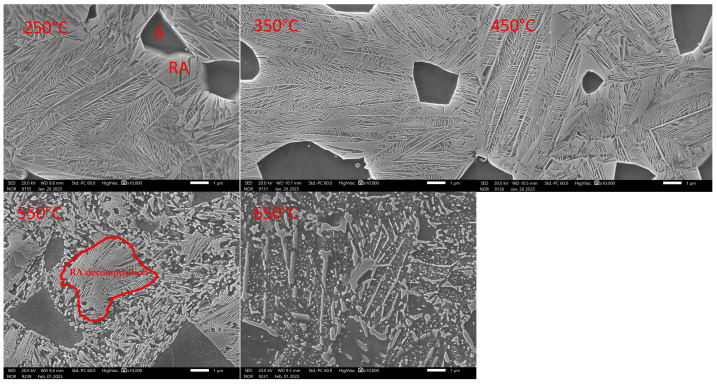
Detail of the microstructures constructed during tempering dilatometric measurements. Heat 7Al Magnification 10,000× in SED observation mode.

**Figure 14 materials-16-03852-f014:**
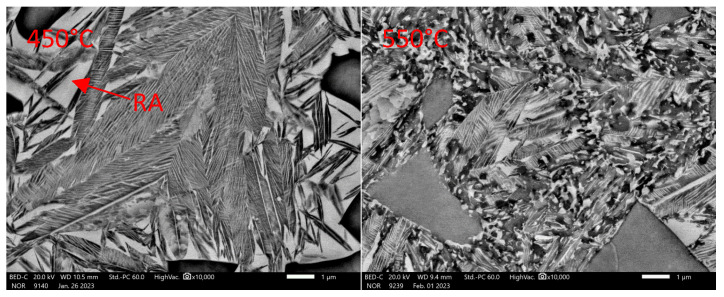
Detail of the microstructures constructed during tempering dilatometric measurements. Heat 7Al Magnification 10,000× in BED observation mode.

**Figure 15 materials-16-03852-f015:**
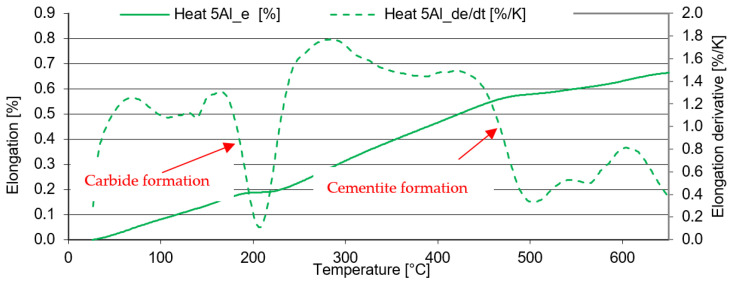
Tempering dilatometric curve—heating at 3 °C/min. Heat 5Al.

**Figure 16 materials-16-03852-f016:**
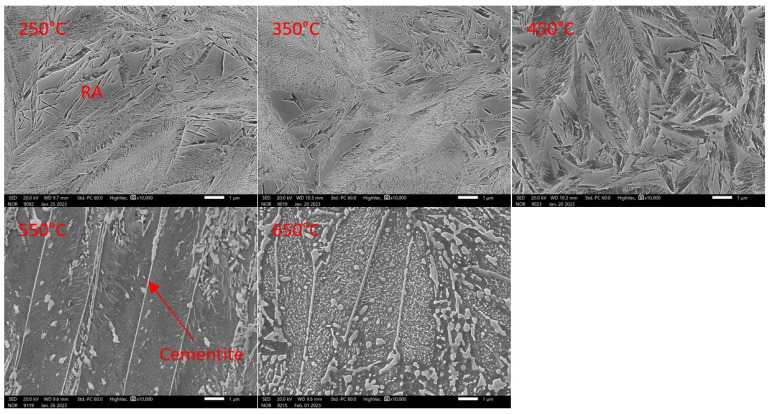
Detail of tempering microstructures formed during dilatometric measurements. Heat 5Al. Magnification 10,000× in SED observation mode.

**Figure 17 materials-16-03852-f017:**
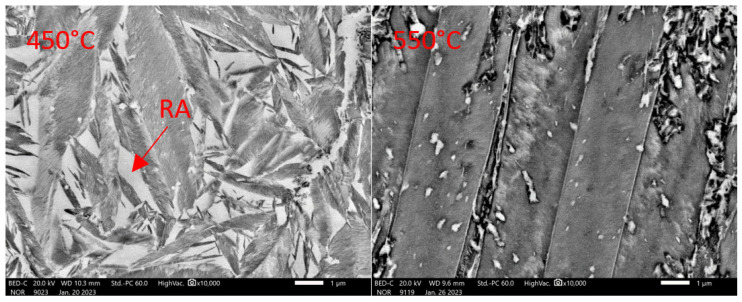
Detail of tempering microstructures formed during dilatometric measurements. Heat 5Al. Magnification 10,000× in BED observation mode.

**Figure 18 materials-16-03852-f018:**
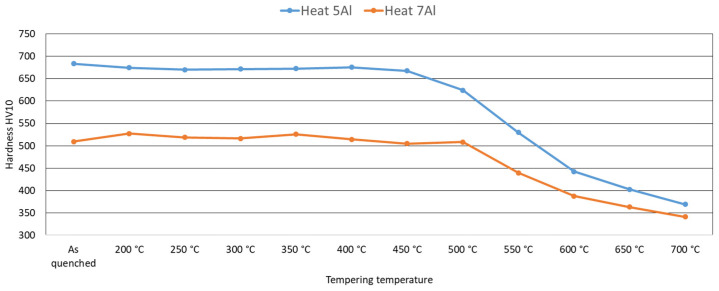
The dependence of hardness on tempering temperature for the heat treatment conditions given in Table 7 and Table 8.

**Figure 19 materials-16-03852-f019:**
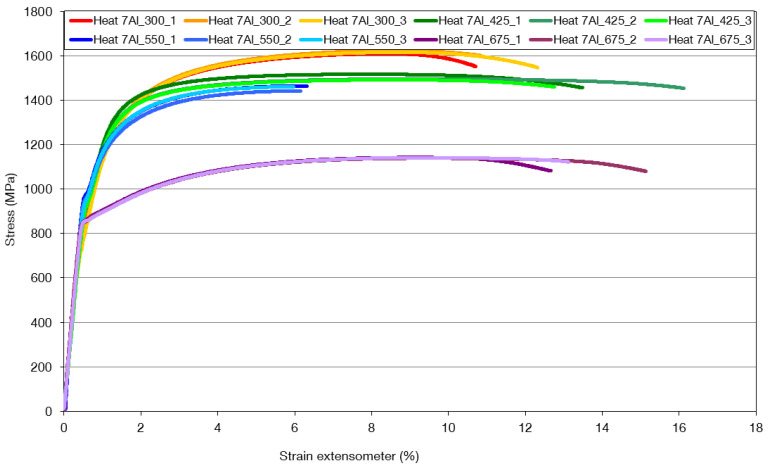
A record of the strength and deformation characteristics of heat 7Al after quenching and tempering to temperatures of 300 °C, 425 °C, 550 °C, and 675 °C.

**Figure 20 materials-16-03852-f020:**
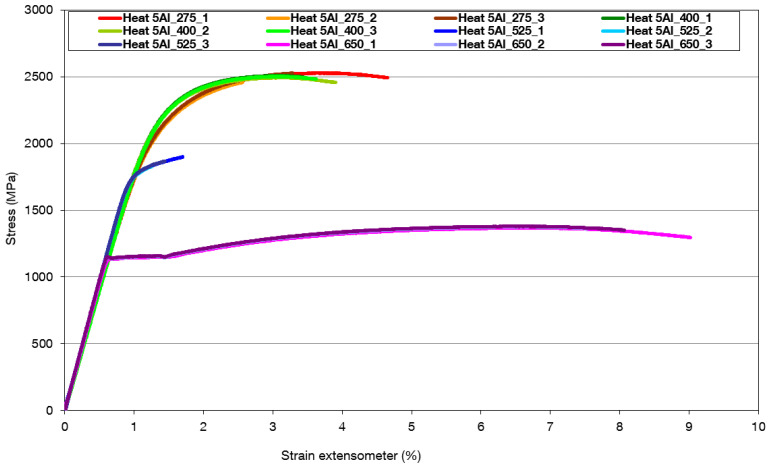
The record of strength and deformation characteristics of heat 5Al (1% C, 5% Al) after quenching and tempering to temperatures of 275 °C, 400 °C, 525 °C, and 650 °C.

**Figure 21 materials-16-03852-f021:**
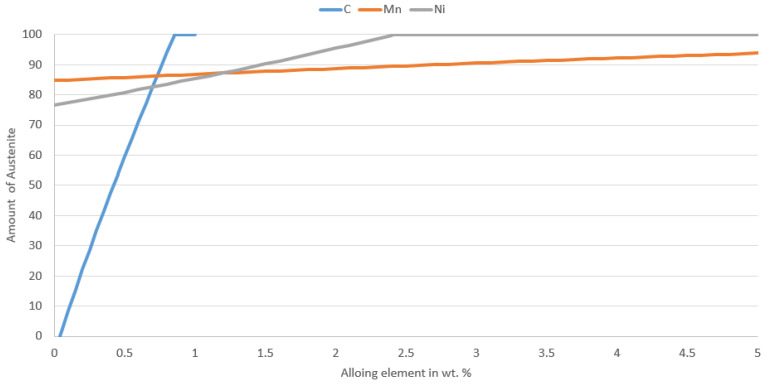
Dependence of austenite content on the alloying element (wt%). Calculation by JMatPro version 12.1.

**Figure 22 materials-16-03852-f022:**
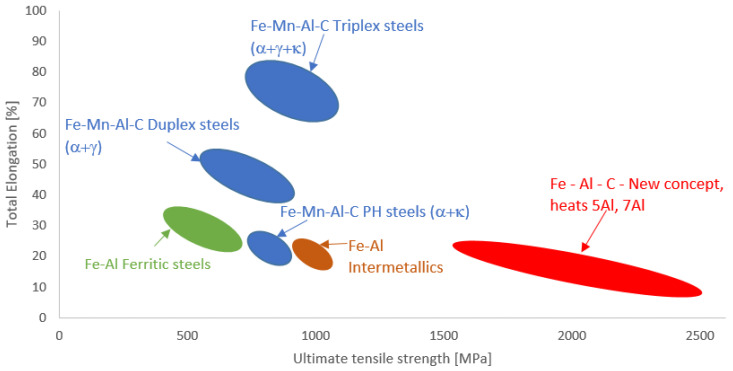
Strength-elongation plot of some low-density steels.

**Table 1 materials-16-03852-t001:** Chemical composition of experimental heats (wt%).

Heat	C	Si	Mn	Cr	Ni	Al	S	P
**7Al (7%Al)**	**0.73**	0.12	0.55	2.02	1.05	**7.06**	0.008	0.01
**5Al (5%Al)**	**1.07**	0.16	0.53	2.03	1.56	**4.97**	0.006	0.008

**Table 2 materials-16-03852-t002:** The proportion of ferrite as a function of temperature Heat 7Al—heating up at 3 °C/min.

Heat_Temperature	Proportion of Ferrite
7Al_930 °C	36
7Al_960 °C	35
7Al_1000 °C	23
7Al_1030 °C	21
7Al_1060 °C	13
7Al_1100 °C	13

**Table 3 materials-16-03852-t003:** Results of the spot analysis of the chemical composition of individual phases. They are all in wt%. Heat 7Al (0.7% C, 7% Al).

Element	Phase	840 °C		Phase	960 °C	
		Av.	St. Dev.		Av.	St. Dev.
Al K	δ	3.9	0.05	δ	3.7	0.17
Cr K		2.0	0.08		2.8	0.20
Mn K		0.8	0.02		0.7	0.12
Fe K		92.2	0.04		91.9	0.48
Ni K		1.1	0.10		0.9	0.12
Al K	κ	6.4	0.3	γ	4.1	0.2
Cr K		3.4	0.4		2.4	0.2
Mn K		1.0	0.1		0.7	0.2
Fe K		88.2	0.5		91.9	0.5
Ni K		0.9	0.1		1.0	0.1

**Table 4 materials-16-03852-t004:** The results of the spot analysis of the chemical composition of individual phases. All values are in wt%.

Phase	Element	810 °C		900 °C	
		Av.	St. Dev.	Av.	St. Dev.
Matrix	Al	2.9	0.34	3.6	0.1
	Cr	2.3	0.49	2.6	0.7
	Mn	0.4	0.31	0.5	0.2
	Fe	92.9	0.35	91.9	0.8
	Ni	1.5	0.05	1.4	0.1
γ	Al	2.7	0.3	2.7	0.11
	Cr	2.4	0.4	3.5	0.76
	Mn	0.7	0.1	0.9	0.0
	Fe	92.8	0.1	91.7	0.53
	Ni	1.4	0.0	1.3	0.14

**Table 5 materials-16-03852-t005:** Experimental parameters to determine the temperature martensite start.

Heat	Austenitisation Temp.	Austenitisation Time	Cooling Rate	
7Al (0.7% C, 7% Al)	930 °C	20 min	50 °C/s	100 °C/s
5Al (1%, 5% Al)	900 °C	20 min	50 °C/s	100 °C/s

**Table 6 materials-16-03852-t006:** Martensite start temperatures for each heat.

Heat	Cooling Rate	
	50 °C/s	100 °C/s
7Al	55 °C ± 4.9 °C	76 °C ± 3.5 °C
5Al	115 °C ± 6.6 °C	131 °C ± 0.7 °C

**Table 7 materials-16-03852-t007:** The heat treatment conditions of specimens from Heat 7Al. All specimens were quenched from the austenitising temperature in oil.

Austenitisation	Tempering										
930 °C	200 °C	250 °C	300 °C	350 °C	400 °C	450 °C	500 °C	550 °C	600 °C	650 °C	700 °C
20 min	120 min										

**Table 8 materials-16-03852-t008:** The heat treatment conditions of specimens from heat 5Al. All specimens were hardened from austenitising steel in oil.

Austenitisation	Tempering										
900 °C	200 °C	250 °C	300 °C	350 °C	400 °C	450 °C	500 °C	550 °C	600 °C	650 °C	700 °C
20 min	120 min										

**Table 9 materials-16-03852-t009:** The heat treatment schedule for tensile test specimens.

Heat	Austenitisation	Quenching	Tempering			
7Al	930 °C/20 min	Oil	300 °C	425 °C	550 °C	675 °C
5Al	900 °C/20 min	Oil	275 °C	400 °C	525 °C	650 °C

**Table 10 materials-16-03852-t010:** Tensile test results at tempering temperatures of 300 °C, 425 °C, 550 °C, and 675 °C. Heat 7Al.

Tempering Temperature	Rp_0.2_ [MPa]	Rm [MPa]	A [%]	E [GPa]
300 °C	919 ± 9	1616 ± 6	10.7 ± 0.4	184 ± 0
425 °C	1017 ± 14	1501 ± 4	14.2 ± 1.4	198 ± 10
550 °C	1026 ± 9	1455 ± 10	6.5 ± 0.1	201 ± 4
675 °C	862 ± 5	1140 ± 6	9.5 ± 0.4	209 ± 5

**Table 11 materials-16-03852-t011:** Tensile test results at tempering temperatures of 275 °C, 400 °C, 525 °C, and 650 °C. Heat 5Al.

Tempering Temperature	Rp_0.2_ [MPa]	Rm [MPa]	A [%]	E [GPa]
275 °C	2078 ± 19	2494 ± 49	2.3 ± 1.1	204 ± 3
400 °C	2193 ± 13	2502 ± 11	2.8 ± 0.4	205 ± 2
525 °C	1802 ± 4	1864 ± 51	0.5 ± 0.3	188 ± 2
650 °C	1141 ± 5	1372 ± 6	8 ± 0	188 ± 5

**Table 12 materials-16-03852-t012:** A table summarising the experimental results.

Heat	Rm [MPa]	A [%]	Aust. T. [°C]	Ms_cooling rate 100°C/s_ [°C]	HV10	ρ [g/cm^3^]
5Al	1372–2502	0.5–8	900	131	369–683	7.18
7Al	1140–1616	6.5–14.2	930	76	341–522	7.08

## Data Availability

The raw data are not publicly available due to ongoing research.

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
