# Peer review of "Mechanical Properties of High Carbon Low-Density Steels"

_materials, 2023, doi:10.3390/ma16103852_

Round 1
Reviewer 1 Report
The paper "Mechanical Properties of High Carbon Low-density Steels" considers the two alloys Fe-5Al-1C and Fe-6Al-1C which, with densities of 7.09 g/cm³ and 7.18 g/cm³ respectively, belong to the low-density steel group compared to conventional steels with densities of around 7.85 g/cm³. Not only the mechanical properties of the alloys are shown in the form of tensile tests at different heat treatment conditions, but also the thermo-physical measurements and recordings of the microstructure after casting are shown.
In the main part of the paper figures, diagrams and tables are embedded in all necessary places. For the large number of results shown, it would also make sense with regard to the chapter "Conclusion" to show a graphical abstract that summarizes the results in a visualized way.
In the evaluated tensile tests in Tables 9 and 10, the evaluation of the Young's modulus of the respective alloys is missing; since this was included in the tensile tests, an evaluation is possible without great effort. Furthermore, it is recommended to draw at least three, but preferably more, specimens for these tensile tests in order to obtain a statistical validation. In the present evaluation, however, only two samples were taken or presented, see figure 19 and 20.
The paper mentions that the ingots were forged, rolled and then the specimens are taken out. However, the microstructure shown afterwards does not specify from which area the specimens were taken (e.g. edge area or center area). Because of the previous forming (microstructure change) and inhomogeneous cooling, there is no homogeneous structure in the rolled plate.
Overall, the paper shows very good and detailed experiments of the two alloys. What is missing, however, are comparisons with other comparable low-density steels, but also with other conventional high-strength steels, and thus the classification of the results shown in the overall context of these alloys and other comparable alloys.
Finally, a few formal minor points are noted below.
In Figure 12, in the legend for the elongation the unit is missing (whereas the elongation derivative has it). The diagrams in Figures 8, 11, 12, 15, 18, 19, 20 and 21 are not consistent with respect to formatting. For example, here the border of the diagram is different, the color of the axes is different, etc. In figure 11, due to missing labels, it is unclear which curves are to be assigned to elongation and which are to be assigned to transformation. In table 2 and 3 there is a letter "K" after each element name, but it is not clear what this letter means.
In line 574 and 578 the headings are not formatted correctly, they should be underlined.
In summary I recommend a major revision of the paper.
-
Reviewer 2 Report
1. The experimental steel samples were smelted by vacuum induction furnace. Top slag, vacuum degree and protective atmosphere should be added.
2. From the metallographic analysis, it was found that the amount of ferrite gradually decreased with increasing temperature. The amount of ferrite should be quantitative analysis. And how about the comparative analysis with the steel 5Al?
3. The steel 7Al reaches an ultimate strength of over 1600 MPa with a ductility of over 10%. Steel 5Al reaches a tensile strength of over 2500 MPa. The effect of Al content on the strength should be explained clearly.
Round 2
Reviewer 2 Report
The responses of authors are very well.